# Probing ultrafast heating and ionization dynamics in solid density plasmas with time-resolved resonant X-ray absorption and emission

Heating and ionization are among the most fundamental processes in relativistic laser–solid interactions; however, their spatiotemporal evolution remains challenging to capture experimentally. Here we present detailed diagnosis of high-intensity laser interactions with wire targets, leveraging the extreme spectral brightness of an X-ray free-electron laser in sub-picosecond time-resolved resonant X-ray emission spectroscopy and absorption imaging. Experimental results are compared with comprehensive simulations using atomic collisional-radiative models, particle-in-cell, and magnetohydrodynamics codes to elucidate the underlying physics. These multi-scale simulations reveal extreme sensitivity of basic plasma parameters with widely used models, such as temperature and ionization depth, which are able to be constrained by incorporating a detailed accounting of laser spatial profiles, pre-plasma conditions, and collisional processes. These results provide new insights into heating and ionization dynamics in the high-energy-density regime relevant to inertial fusion energy research, both as an experimental platform for accessing theoretically challenging conditions and as a benchmark for improving models of high-power laser–plasma interactions.

Upon irradiation of a solid target by an ultra-short relativistic laser pulse, electrons within the interaction region are promptly ionized and accelerated to kinetic energies of up to several tens of MeV. The non-thermal energetic electrons then transport into the bulk of the target, initiating abundant subsequent phenomena, including the generation of resistive return currents and extreme electromagnetic fields, the growth of plasma instabilities, Bremsstrahlung and line radiation emission, as well as ion acceleration. Ultrafast heating and ionization are among the most fundamental processes, determining the key plasma parameters such as current density, temperature, opacity and resistivity that in turn govern the electron transport dynamics[1]. Understanding the complex dynamics is crucial for advancing the applications of laser plasma-based accelerators[2], creation of high-energy-density matter[3] and laser-driven fusion

energy[4,5]. To date, predictive insights into the heating and ionization physics of solid-density plasmas primarily rely on large-scale numerical plasma simulations[6–9], as experimental efforts remain a major challenge due to the limited accessibility of over-critical density and the inadequate spatiotemporal resolution of conventional diagnostics. X-ray emission spectroscopy (XES) serves as a powerful diagnostic to probe the plasma temperature and density, typically through analysis of the intensity and ratios of characteristic X-ray lines at specific transition energies, such as $K_\alpha$, $K_\beta$, and $He_\alpha$ emissions[10–13]. Integrating an X-ray streak camera with XES enables time-resolved spectroscopy. However, its temporal resolution is currently limited to ~1 ps[14], which is insufficient to resolve the ultrafast dynamics occurring during the intra-laser-matter interaction phase. Monochromatic Bragg crystal imaging has also been

✉e-mail: lingen.huang@hzdr.de

developed to provide spatially resolved ionization maps. Yet, its spatial resolution is typically limited to scales larger than ~10 μm[15,16].

Over the last decade, the advent of ultra-short, high-brilliance X-ray free electron lasers (XFELs) with an extremely high photon number per pulse (in the order of $10^{12}$), ultra-short duration (tens of fs) and narrow spectral bandwidth (~20 eV in self-amplified spontaneous emission mode or ~1 eV in seeding mode) exhibiting nearly full transverse coherence[17], provides unprecedented opportunities to probe the transient, non-thermal, solid-density plasmas under extreme conditions driven by high-intensity optical lasers. XFELs enable visualization of plasma dynamics with temporal resolution on the order of tens of fs and spatial resolution ranging from a few nm to sub-μm scales simultaneously. Recent pioneering work at the existing large-scale facilities equipped with both optical high-power laser and XFEL has demonstrated the exceptional capability to observe, in detail, phenomena such as solid-density plasma expansion[18,19], kinetic instabilities[20,21], and cylindrical compression[19,22]. Experimental measurements of plasma opacity and ionization have also been proposed and carried out using resonant small-angle X-ray scattering[23,24]. Most recently, the novel experimental pump-probe platforms have been employed to investigate the heating and ionization dynamics of high intensity laser irradiated Cu foils, via X-ray transmission imaging near the Cu K-edge[25]. However, due to the stochastic nature of the XFEL beam—particularly its spatial and temporal jitter—the resulting transmission imaging maps may suffer from significant uncertainties. In this work, we explore the heating and ionization dynamics in high-power laser-driven Cu wires using time-resolved resonant X-ray emission spectroscopy, combined with simultaneous X-ray absorption imaging. Compared with the Cu foils used in the previous experiments[25], the cylindrical target geometry eases the alignment and overlap of the optical laser and XFEL on the target, as well as significantly localizing heating effects of electron collisions with reduced target dimensions. Furthermore, the initial uncertainties in laser-target coupling caused by beam jitter can be quantified through measurements of the K-satellite and He$_\alpha$ emission yields, and the transmitted X-ray imaging profiles. These diagnostic advantages make the wire target particularly well-suited for maximizing the capabilities of high-energy-density (HED) facilities.

In this experiment, the XFEL photon energy was tuned to 8.2 keV to be resonant with an electron transition between two bound states of highly charged nitrogen-like Cu$^{22+}$ ions generated by the high-power optical laser. When the specific charged state is populated, the XFEL photons resonantly excite a K-shell bound electron to a higher L-shell state with a corresponding increase in plasma opacity, resulting in enhanced resonant emission through de-excitation. By varying the time delay between the XFEL and optical laser driver, we observed a clear rise in the resonant X-ray emission yield—proportional to the Cu$^{22+}$ ion population—that peaks at ~2.5 ps after the laser pulse maximum and decays over a timescale of up to 10 ps. Furthermore, a distinct correlation between resonant emission and attenuation of X-ray beam is observed, inferring the consistent underlying heating, ionization and recombination dynamics. Such simultaneous measurements isolate the contribution from a single charge state with sub-picosecond time resolution, enabling quantitative tracking of both the evolution of a specific ionic population and the resonant opacity at solid density. This observed correlation implies that the resonant features are confined to a small spatial scale length near the front surface, on the order of μm. To our best knowledge, no previous diagnostic has provided time-resolved access to both charge-state-selective resonant absorption and emission in a highly transient, solid-density plasma. In addition, neighboring satellite emissions in the vicinity of resonant XFEL photon energy are clearly detected. These features reflect complex atomic processes in which, besides the dominant on-resonance radiative decay, Auger decay and collisional relaxation of excited ions can also contribute to the de-excitation

following X-ray resonant pumping, thereby altering the electronic configuration and emitted spectrum. Such off-resonance emission has also been reported in previous XFEL-only studies[26]; however, it is now observed in a laser-generated hot dense plasma, where non-thermal electron populations with MeV energies can significantly modify collisional rates. Here, the off-resonance emission line intensities on both sides of the XFEL photon energy are similar, suggesting that the collisional ionization and recombination rates are comparable and within an order of magnitude of the radiative K-L transition rate. To unveil the complex dynamics, comprehensive simulations using atomic collisional-radiative, particle-in-cell (PIC) and magneto-hydrodynamics (MHD) codes were performed, incorporating both local thermodynamic equilibrium (LTE) and non-LTE (NLTE) ionization models. Comparison with experimental data highlights the sensitivity of the predictive capability of the simulation framework to the initial laser-target interaction parameters and modeling choices. Specifically, a realistic peak laser intensity, based on measured laser spatial profiles, together with appropriate preplasma conditions, is found to be essential for state-of-the-art simulations incorporating non-equilibrium collisional processes to reproduce the experimentally observed plasma heating and ionization dynamics.

## Results and discussion
### Experimental overview

The pump-probe experiment was performed at the HED-HiBEF instrument located at the European XFEL using the ultra-intense optical laser ReLaX to create the solid-density plasmas. ReLaX is a Ti:sapphire-based facility delivering 3 J, 30 fs pulses at a 10 Hz repetition rate[17,27]. Based on the measured laser intensity profile, the actual focus deviates from a perfect Gaussian, with only 44% of the energy in the central focus (9% relative standard deviation) and FWHM sizes of 3.9 μm × 3.6 μm (12% and 11% relative standard deviations) in the horizontal and vertical directions. Fluctuations in laser energy and pulse duration are negligible. Combining all of these factors yields a peak intensity of $2.5 \times 10^{20}$ W/cm$^2$, with a relative standard fluctuation below 25%, upon normal-incidence irradiation of 10 μm-diameter Cu wires. The XFEL beam generated by the principle of self-amplified spontaneous emission (SASE) with the energy of ~1.5 mJ, FWHM pulse duration of ~25 fs, FWHM spot size of ~10 μm and photon energy centered at 8.2 keV with a FWHM bandwidth of ~18 eV was used to probe the solid-density plasmas. The heating and ionization dynamics were investigated primarily through time-resolved resonant X-ray emission spectroscopy (RXES) in backward direction, complemented by simultaneous X-ray absorption imaging acquired via a scintillator screen coupled to an Andor Zyla CMOS camera. Spectra were recorded using a von Hámos spectrometer equipped with a highly annealed pyrolytic graphite crystal[28]. A schematic illustration of the experimental setup, along with detailed experimental conditions, is described in the Methods section and our previous publication[19].

### Atomic simulation

In order to understand the mechanism of atomic processes in resonant X-ray emission, we first conducted simulations with the widely used atomic collisional-radiative code SCFLY, which solves the rate equations governing ionization balance by accounting for various processes such as collisional ionization, radiative and three-body recombination, excitation, de-excitation, Auger ionization, and dielectronic recombination[29]. Figure 1a shows the calculated opacity spectra of solid-density Cu plasmas with a thickness of 10 μm, over a temperature range from 150 eV to 1 keV. A distribution of multiple satellite lines with slightly overlapping peaks can be observed. Each peak corresponds to a bound-bound transition, with its transition energy depending on the number of other electrons in L-shell and thus Cu charge state. In the case of plasma temperature at 150 eV, only doublet peaks of $K_{\alpha 1}$ and $K_{\alpha 2}$ of Cu at 8.047 keV, 8.027 keV are clearly

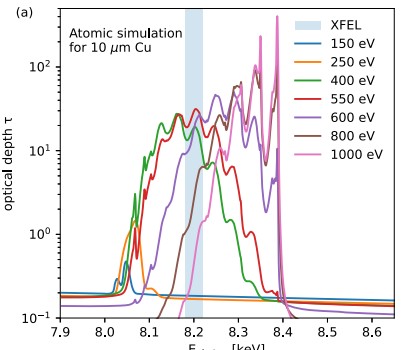
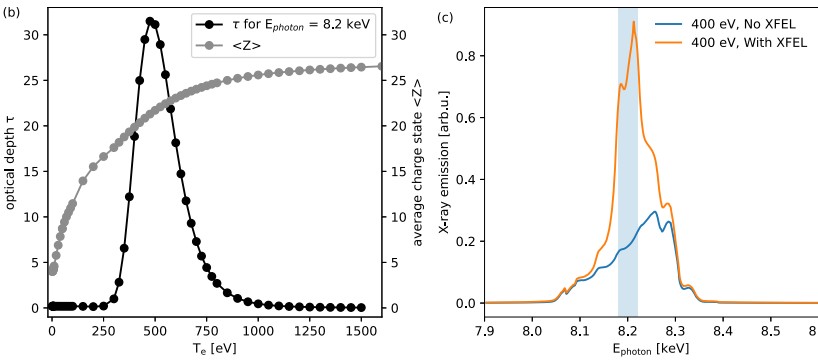

**Fig. 1 | Atomic simulation of plasma opacity and X-ray emission spectra.**
**a** Optical depth of a solid Cu plasma with 10 μm thickness at different temperatures calculated by the atomic code SCFLY over a broad photon energy range covering bound-bound and bound-free transitions for the hot dense plasma. Each peak corresponds to a specific K-L bound-bound transition of the Cu ions. **b** Temperature dependence of Cu's optical depth at the resonant photon energy of $E_{photon} = 8.2$ keV, along with its corresponding averaged charge state. **c** Comparison of simulated X-ray emission spectra with and without XFEL irradiation at a plasma temperature of 400 eV. The light blue shaded region denotes the full spectral bandwidth of the XFEL beam centered at 8.2 keV.

visible. As plasma temperature increases, the average charge state of the ions rises, as shown in Fig. 1b. This leads to a shift of the spectral peak positions toward higher photon energies, due to reduced electron screening of the nuclear Coulomb potential. Consequently, the transition energies increase, opening the possibility of using resonant X-ray absorption as a diagnostic for probing atomic processes in the plasma. In this experiment, the XFEL photon energy was tuned to 8.2 keV, which is predominantly in resonance with the bound-bound transition of the highly charged nitrogen-like $Cu^{22+}$ ion in plasmas with a specific charge state. The correlation between plasma temperature, charge state distribution, and opacity at this selected photon energy ($E_{photon} = 8.2$ keV) is illustrated in Fig. 1b. When the $Cu^{22+}$ ion population reaches its maximum—corresponding to a plasma temperature of ~500 eV—the plasma opacity also peaks, and the probability of XFEL resonance absorption is at its highest. As the solid-density plasma continues to heat beyond this point, further ionization results in charge states exceeding $Cu^{22+}$. With fewer bound electrons available to support the resonant transition, both plasma opacity and XFEL absorption decrease. In the hot, dense plasmas driven by the optical laser, X-ray emission from the studied transition can occur spontaneously. When an intense XFEL with photon energy $E_{photon} = 8.2$ keV is applied, the innermost K-shell electrons are resonantly excited by X-ray photons to higher-energy L-shell states. The excited electrons then mainly decay through radiative process, leading to enhanced X-ray fluorescence yield at the resonant energy as shown in Fig. 1c. Furthermore, enhanced neighboring emissions in the energy range beyond the XFEL bandwidth (FWHM = 18 eV) on either side of the resonance are observed, suggesting that Auger decay, collisional ionization, and recombination via dielectronic and three-body processes also contribute to the total emission, in addition to the dominant on-resonance radiative decay, as also reported in previous XFEL-only studies[26]. It is important to note that the atomic simulations performed here assume uniform plasma conditions. In contrast, the solid-density plasma generated by a high-power laser is typically non-equilibrium and exhibits significant temperature and density gradients. As a result, the realistic X-ray emission spectrum is a composite of contributions from regions with varying plasma conditions, as illustrated in Fig. 2.

## Diagnosis of heating and ionization dynamics

Figure 2a shows time-resolved X-ray resonant emission spectra from selected shots, measured at pump-probe delays from 1 ps before to 10 ps after the peak laser pulse, with 0.5 ps temporal resolution. These shots were selected based on consistently elevated $He_\alpha$ emission yields and similar transmitted X-ray imaging profiles, which indicate comparable initial laser-target coupling and XFEL probing conditions. The

justification criteria for shot selection are detailed in the Methods section. The prominent $K_\alpha$ doublet lines at 8.02 and 8.04 keV and $He_\alpha$ transitions between 8.28 and 8.42 keV are observed in all the shots. These transitions have no interference with the XFEL probe; their X-ray emission is time-integrated during the entire laser-solid interaction. The similarity in both spectral shape and absolute yield of the $K_\alpha$ and $He_\alpha$ emission lines further confirms consistent initial conditions across the selected shots. The measured number of $K_\alpha$ photons is ~$(8.0 \pm 1.5) \times 10^{10}$, corresponding to a total emitted energy of ~1.36 mJ assuming isotropic emission, corresponding to the conversion efficiency of ~$4.3 \times 10^{-4}$ relative to the ReLaX laser energy. Prior to the peak pulse arrival, the spectra show that the spontaneous X-ray fluorescence within the energy window covered by the XFEL bandwidth is weak and insufficient to surpass the self-emission background, and thus is buried in the continuum radiation due to the rapid heating. After the peak intensity irradiating on the wires, a clear rise-and-fall temporal evolution of stimulated X-ray emission yield is exhibited within the XFEL bandwidth. Specifically, the resonant X-ray emission yield becomes visible at ~0.5 ps, rapidly ascends to its maximum at ~2.5 ps, and then decays over a timescale of up to 10 ps till it vanishes in the background. The distinct feature is imprinted on the population history of the resonant bound-bound transition of the selected highly charged $Cu^{22+}$ ion, and thus is directly correlated to the dynamics of plasma heating, ionization, and recombination. According to the SCFLY simulation shown above, the measured temporal evolution of the resonant emission suggests that the temperature within the solid-density plasma remains above 500 eV for up to 10 ps. It is worth noting that the same physical principle has been successfully applied to generate highly coherent, attosecond-duration atomic X-ray lasers, where atoms are heated and pumped by intense XFEL pulses[30–32].

Figure 2b shows the time-resolved resonant X-ray emission yield alongside the corresponding XFEL transmission. A clear inverse correlation between the two is observed, as further presented in Fig. 2c, indicating consistent underlying processes of plasma heating, ionization, and recombination. The observed correlation between the resonant X-ray emission yield and the XFEL absorption in Fig. 2c indicates the small scale length of the resonant features, as otherwise absorption becomes saturated, while backward resonant emission yield is dominated by the first absorption length thickness. In this way having both resonant backscatter (sensitive to the x-ray absorption depth) and transmission (bulk) are complementary diagnostics yielding additional information from their combination. For the resonant charge state corresponding to peak absorption at the XFEL photon energy, the effective absorption length is on the order of one micrometer, $O(\mu m)$. This implies that, for linear proportionality, the projected thickness must be smaller than this value, indicating that the ionization is highly

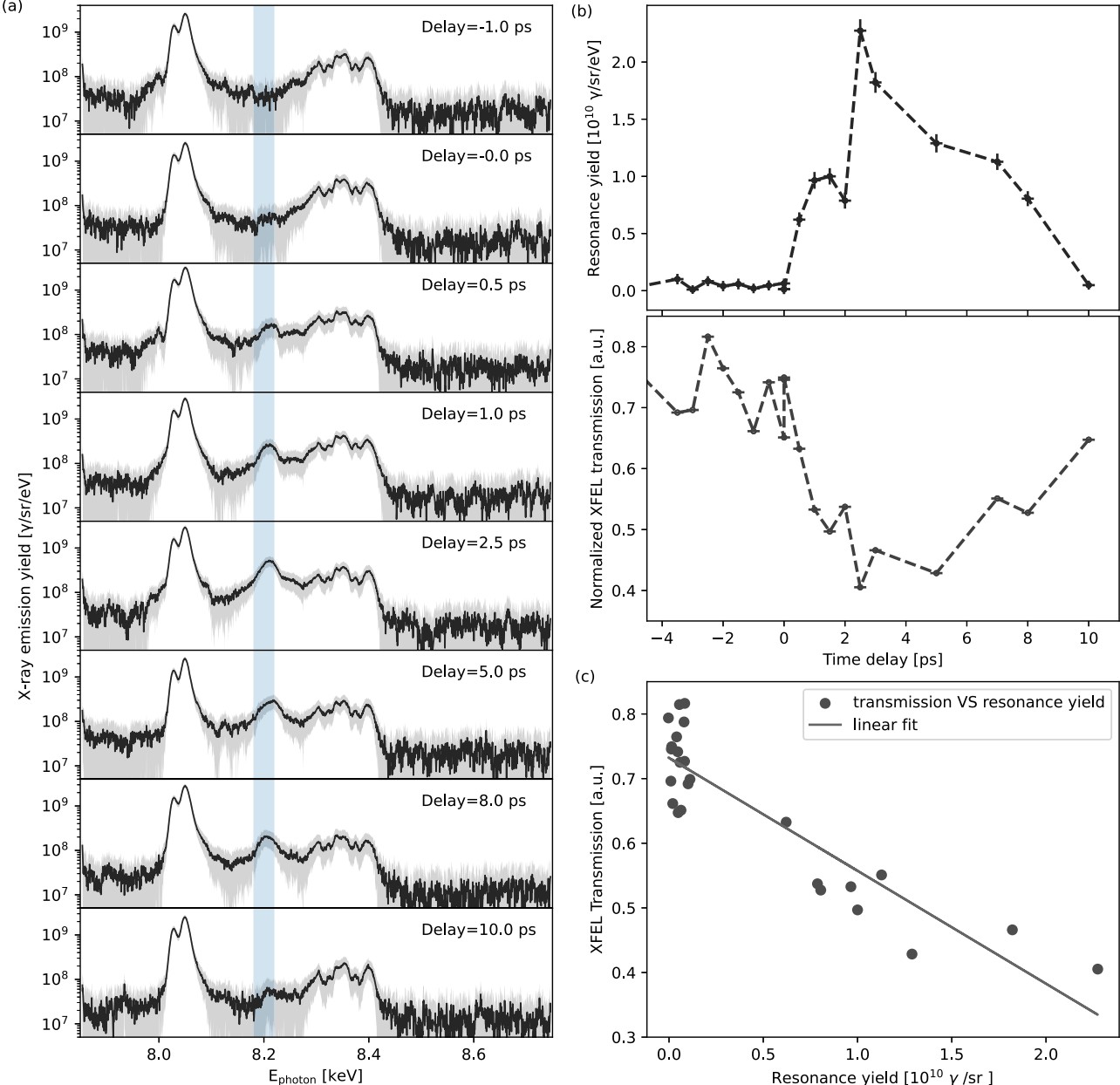

**Fig. 2 | Experimental measurements of resonant X-ray emission yield and XFEL transmission. a** Experimentally measured time-resolved X-ray emission spectra covering the pump-probe delay ranging from −1 ps to 10 ps for a few selected shots. The light blue shaded region denotes the full spectral bandwidth of the XFEL beam centered at 8.2 keV. **b** Temporal evolution of resonant X-ray emission yield in the region of XFEL photon energy and normalized XFEL transmission. The stochastic fluctuation in XFEL transmission is primarily caused by X-ray spatial jitter, particularly evident at negative delays. **c** Correlation of the resonant X-ray emission yield and the XFEL transmission. The error bar of the pump-probe time delays assumes the relative timing uncertainty to be 200 fs for all the hot shots. The error of the spectrum is multiplied with a factor of 5 to make it visible on the figure. The background subtraction and selection of good shots are detailed in the Methods section.

localized near the front surface, rather than producing a uniform charge state. Furthermore, although the features are smaller than the XFEL focal size, variations in direct beam overlap due to beam jitter introduce only minor fluctuations in the total absorption signal. Forward X-ray propagation simulations show that the absorption variation is <0.1 assuming ± 5 μm jitter, which is consistent with the experimental data shown in Fig. 2b at negative delays. Consequently, changes in the resonant absorption remain clearly distinguishable when the effects of beam jitter are taken into account. Thus, while resonant emission serves as an ionization diagnostic and is extensible to thicker or layered targets containing a suitable resonance due to its spectral sensitivity, the observed correlation in resonant absorption can be

used here to infer an approximate scale length even without direct imaging. In particular, in the case of the strongest XFEL stimulation at 2.5 ps, the resonant X-ray yield reaches maxima at $2.27 \times 10^{10}$ photons/sr, which corresponds to $(0.37 \pm 0.02)$ mJ over the full $4\pi$ solid angle. It is ~$(47 \pm 3)\%$ of the XFEL pulse energy measured with ~0.787 mJ. For comparison, the attenuation length of cold solid Cu at the photon energy of 8.2 keV is 22.73 μm according to the Henke Tables[33]. For an XFEL beam with a ~10 μm field of view centered on the 10 μm-diameter cold Cu wire, the total XFEL absorption is calculated to be 22%. This is equivalent to ~46% of the absorption measured in the hot dense plasma at 2.5 ps, consistent with the increased attenuation observed in the resonant case and the opacity predictions shown in Fig. 1, as well as

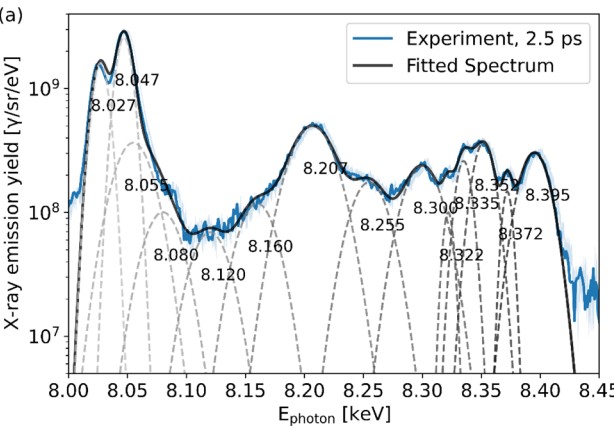

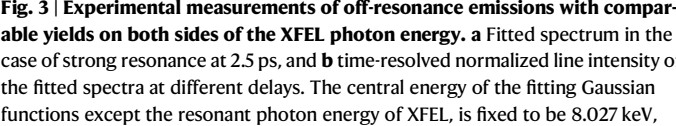

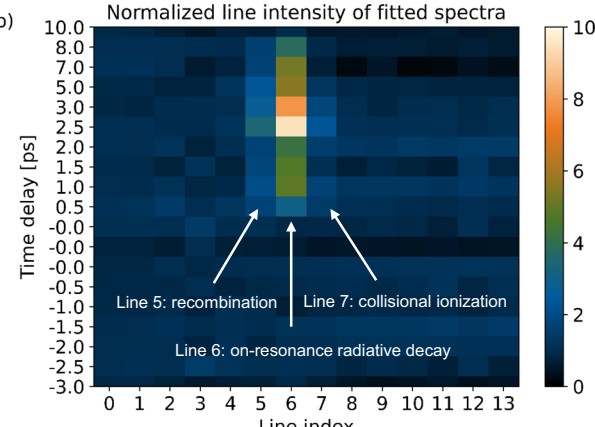

**Fig. 3 | Experimental measurements of off-resonance emissions with comparable yields on both sides of the XFEL photon energy. a** Fitted spectrum in the case of strong resonance at 2.5 ps, and **b** time-resolved normalized line intensity of the fitted spectra at different delays. The central energy of the fitting Gaussian functions except the resonant photon energy of XFEL, is fixed to be 8.027 keV,

8.047 keV, 8.055 keV, 8.08 keV, 8.12 keV, 8.16 keV, -8.2 keV, 8.255 keV, 8.3 keV, 8.322 keV, 8.335 keV, 8.352 keV, 8.372 keV and 8.395 keV, labeled with line indices from 0 to 13, respectively. Each index corresponds to a specific bound-bound transition that emits at a characteristic energy.

the correlation shown in Fig. 2b. Since the radiative decay rate is proportional to the ion number density[29], the measured resonant emission yield directly reflects the population of the selected $Cu^{22+}$ ion, and thus ionization and recombination evolution. It is worth noting that the duration of the resonant X-ray emission lasts significantly longer in the Cu wires than in the foil case, as observed in an ongoing study, where the emission decays completely at ~3 ps. The slower evolution and enhanced emission observed in wires is attributed to lower heat dissipation caused by stronger refluxing and reduced spatial spreading of the surface-confined hot electrons[34].

In the hot, dense plasma state created by the optical relativistic laser pulse, the electronic configurations of Cu ions with K-shell vacancies—pumped by resonant XFEL radiation—can become highly complex. Each spectrum peak shown in Fig. 2a is contributed by an average of many different configurations of Cu ions with different statistical weights and slightly different K-L transition energies. In particular, the resonantly excited L-shell electrons could be relaxed by the channels of radiation decay, Auger decay or collisions. The first case will result in the resonant emission at the same wavelength as the incident XFEL beam, while the other two channels will re-distribute the electronic configuration and thus will result in the K-satellite emission of neighboring states, as discussed in Fig. 1c. To analyze these features, the experimental spectra were fitted by a set of 14 gaussian functions, corresponding to the distribution of each individual satellite emission between $K_\alpha$ and $He_\alpha$. The central energy of the gaussian functions, except the resonant photon energy with XFEL, are fixed as inferred from the SCFLY atomic simulations shown in Fig. 1a and the XFEL-only heating experimental data in well-defined conditions[35], namely, 8.027 keV, 8.047 keV, 8.055 keV, 8.08 keV, 8.12 keV, 8.16 keV, -8.2 keV, 8.255 keV, 8.3 keV, 8.322 keV, 8.335 keV, 8.352 keV, 8.372 keV, and 8.395 keV, labeled with line indices from 0 to 13, respectively. Each index identifies a specific bound-bound transition associated with emission at a characteristic energy. Figure 3a shows an example of a fitted spectrum at 2.5 ps delay, corresponding to the peak of resonant emission yield. To mitigate the uncertainties of initial laser-wire coupling conditions, the intensity of each fitted line is furthermore normalized to the mean value of the shots with negative delays, that are analogous to the laser-only shots since the resonant channel is not opened yet, as already seen in Fig. 2a. Figure 3b shows the normalized line intensity of the fitted spectra at different delays. A prominent line (line 6) at ~8.2 keV is clearly observed, attributed to dominant on-resonance radiative decay. In addition, off-resonance lines on either

side—at 8.16 keV (line 5) and 8.255 keV (line 7)—within an energy window of ~100 eV, are simultaneously pumped by the XFEL field, while the evolution of other lines is not distinct. Since the FWHM bandwidth of XFEL beam is only ~18 eV, which is not broad enough to drive the neighboring satellites, the off-resonance emissions are attributed to the Auger decay, collisional ionization (increasing the charge state), as well as recombination (decreasing the charge state) through dielectronic and three-body processes, which modify the L-shell electron occupancy between resonant excitation and subsequent emission. The ratios of line intensities can therefore be used to measure collisional rates (ionization and recombination) relative to the radiative rate. Here, the off-resonance emission line intensities on both sides of the XFEL photon energy are similar, suggesting comparable ionization and recombination rates, though they are a few times lower than the radiative decay rate. The observed neighboring satellite emissions near the resonant XFEL photon energy are consistent with the SCFLY simulation shown in Fig. 1c. A similar feature was reported in an ultrathin warm dense plasma (~54 nm) isochorically heated by XFEL pulses[26]. In this work, such on- and off-resonance features are observed in a laser-generated hot dense plasma (~10 μm thick) exhibiting steep temperature and density gradients, where nonthermal electron populations with MeV energies can strongly alter collisional rates. In conjunction with time-resolved spectroscopic measurements, X-ray absorption imaging can be further used to probe the spatial distribution of ionization within plasmas, as recently demonstrated in short-pulse laser-driven warm/hot dense matter experiments[25].

## Kinetic and MHD simulations

To further reveal the ultrafast heating and ionization dynamics, two-dimensional (2D) PIC and MHD simulations were performed under realistic laser and target conditions. The PIC simulations employ both an LTE-based pressure ionization model and an NLTE-based impact ionization model implemented within the PICLS code[36]. It is noticed that the LTE model implicitly accounts for recombination via an empirical Thomas-Fermi equation of state (EOS) that constrains the cell-averaged ion charge, while the NLTE model uses a probabilistic Monte Carlo approach to compute the impact ionization rate but neglects recombination. The treatment of the ionization models in the code is detailed in previous theoretical work[8,37]. Additionally, plasma heating is modeled using a relativistic binary collision operator[36] in all the cases. The PIC simulations summarized in Fig. 4 were performed with a lower-bound peak intensity within the standard deviation based

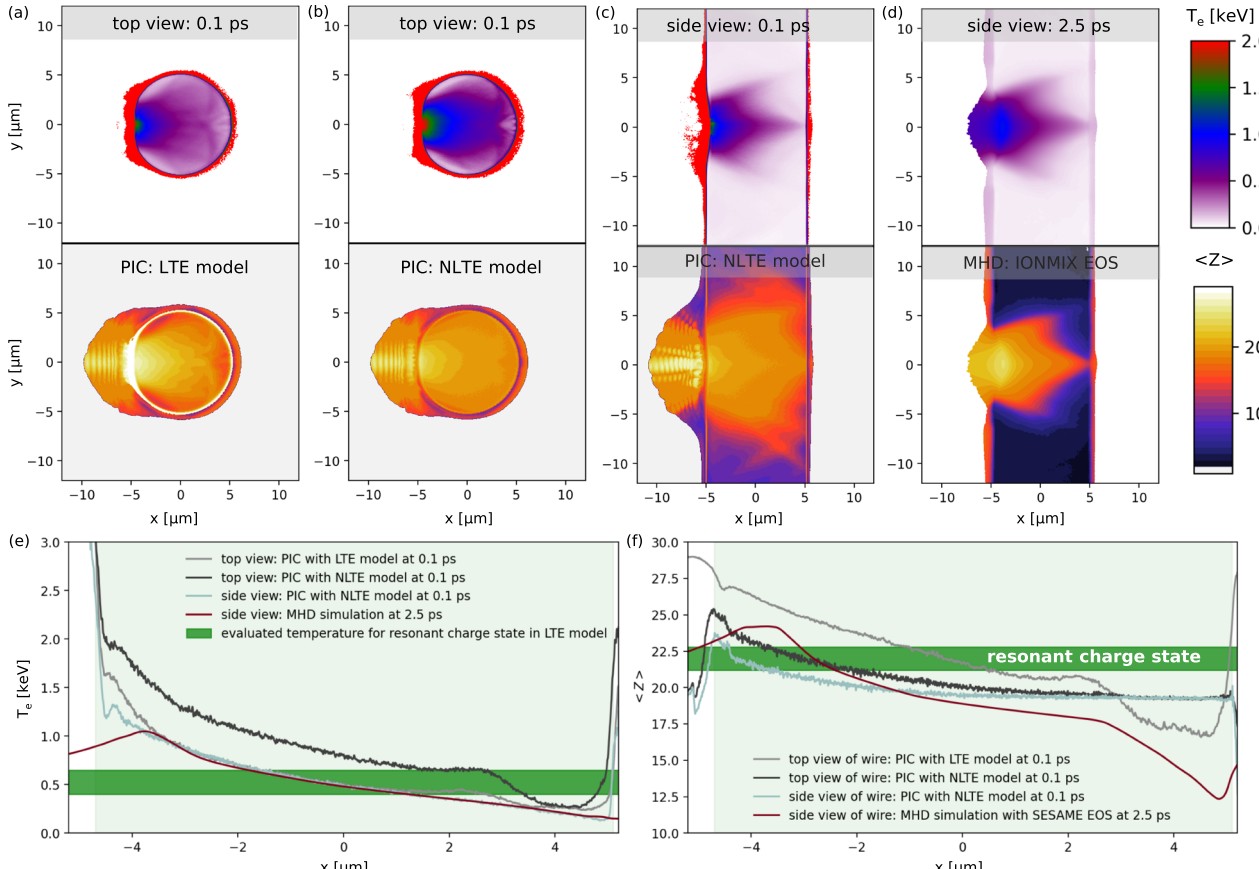

**Fig. 4 | Hybrid kinetic and MHD simulations of plasma heating and ionization dynamics.** Spatial distribution of electron temperature and ion charge state obtained from 2D PIC simulations in the case of top view of a Cu wire with 10 μm thickness, using **a** the LTE-based Thomas-Fermi and **b** the NLTE-based direct impact ionization models, both at 100 fs after peak laser intensity irradiation. Corresponding 2D PIC and MHD simulations for the side view of the Cu wire using the NLTE ionization model are shown at **c** 100 fs and **d** 2.5 ps, respectively. Longitudinal lineouts of the plasma temperature (**e**) and average ion charge state (**f**) of the corresponding 2D simulation maps, extracted from the initial target region indicated by the light green area. The light violet regions indicate the probed resonant charge state and its corresponding evaluated plasma temperature. The parameters of the PIC simulation are described in the Methods section.

on the measured spatial laser profiles, i.e., $1.875 \times 10^{20}$ W/cm², that accounts for the non-negligible fraction of energy distributed outside the central focus region. Additionally, the simulations employed a preplasma profile predicted by MHD simulations using the FLASH code with the IONMIX EOS library[38], which shows good agreement with experimental measurements obtained via nanometer-sensitive small-angle X-ray scattering diagnostics in our recent work[19]. Details of the simulation setup are provided in the Methods section. Firstly, we examine the comparison of plasma temperature and ionization for the LTE and NLTE ionization models using a circular geometry that represents a top-down view of a wire target at 0.1 ps shown in Fig. 4a, b. In both cases, the front surface temperature at the interaction point is heated up to ~3 keV and experiences a rapid decline within the target. In the bulk, although the NLTE model predicts temperatures that are a few hundred eV higher, the resulting ionization depth for the highly ionized resonant charge state $Cu^{22+}$ is much smaller—~2 μm compared with 5 μm in the LTE model. The lower ionization is consistent with the higher temperature observed in NLTE case, since less thermal energy is extracted from the bulk plasma for ionization than in the LTE case. High-intensity, ultrafast laser-generated solid-density plasmas often deviate from LTE conditions characterized by a Maxwellian particle energy distribution. The NLTE ionization model is more accurate in this regime, as it accounts for arbitrary electron energy distributions, including relativistic tails. As shown in Fig. 4f, the NLTE ionization model provides a closer match to the experimental data, which reveals

localized surface ionization as discussed in Fig. 2. These comparisons demonstrate the critical role of non-equilibrium collisional processes in improving the predictive capability of the simulations. Furthermore, we conducted additional 2D PIC simulations employing the NLTE ionization model in a planar configuration, depicting the side view of the wire target as plotted in Fig. 4c. It is evident that both the surface and in-depth heating and ionization levels caused by the electron transport are notably reduced in comparison to the previous circular scenarios. This reduction occurs because the hot electrons spread laterally along the planar target surface nearly at the speed of light, being primarily constrained by the longitudinal sheath fields. In contrast, within the context of a mass-limited circular geometry, the hot electrons are effectively confined or reflected by strong sheath fields encompassing its entire surface. In this case, the ionization depth for the resonant charge state $Cu^{22+}$ is further reduced to ~1.5 μm, in excellent agreement with the experimentally measured effective absorption length discussed earlier. Limited by computational power, we are only able to run the PIC simulations with realistic solid density up to 100 fs when the laser intensity drops far below the relativistic intensity. To extend the temporal analysis, we have run an MHD simulation using the PIC predicted distributions of electron and ion temperatures to model the plasma adiabatic cooling through the hydrodynamic expansion and thermal equilibration. As shown in Fig. 4d, 2.5 ps after the laser−solid interaction, the surface electron temperature decreases to ~1 keV, and the region corresponding to the

resonant charge state $Cu^{22+}$ extends to a depth of ~2.5 μm. This region remains localized near the surface and is therefore in satisfactory agreement with the measurements, although the MHD simulations implicitly assume equilibrium conditions. It is noted that, in order to best reproduce the key experimentally measured parameters—such as the heating and ionization depth—it is essential to account for the realistic peak laser intensity based on the measured spatial laser profile, as well as an appropriate preplasma profile. These are the most critical initial conditions for the PIC simulations in constraining the basic plasma parameters and are discussed in detail in the Methods section "Effect of Peak Laser Intensity and Preplasma Profile".

## Discussion

In this study, the ultrafast heating and ionization dynamics in a hot dense plasma were investigated using a novel platform in which an ultra-short, relativistic optical laser driver was combined with an XFEL probe. It enables sub-picosecond time resolution over an array of complementary diagnostics, including resonant emission spectroscopy and simultaneous X-ray absorption imaging as sensitive charge state population diagnostics. The x-ray emission yield in the energy window resonant with the XFEL beam at 8.2 keV was observed at ~0.5 ps, rapidly peaks at ~2.5 ps, and then decays gradually until disappearing at ~10 ps after the laser irradiation of the Cu wires. The distinct rise-and-fall temporal profile of the on-resonance emission yield reflects the evolving population of highly ionized $Cu^{22+}$ ions, indicating ongoing ionization and recombination processes while the solid-density plasma temperature remains above 500 eV up to ~10 ps after the laser target interaction. A clear linear correlation between the yield of resonant x-ray emission and the XFEL attenuation at different pump-probe delays was observed, consistent with atomic collisional-radiative simulations. Furthermore, clearly resolved neighboring off-resonance features were also observed. It suggests that, in addition to the dominant radiative decay, the excited ions stimulated by the XFEL can also undergo relaxation via Auger and collisional processes (ionization and recombination). Comparable off-resonance line intensities on both sides of the XFEL photon energy suggest similar ionization and recombination rates. The experimental results were compared with comprehensive simulations using PIC and MHD codes to elucidate the underlying physics. The hybrid MHD-PIC-MHD simulation framework, which incorporates a realistic peak laser intensity based on the measured spatial laser profile together with an appropriate MHD-predicted preplasma profile, is able to reproduce key experimentally measured parameters, such as the heating and ionization depths when the NLTE model is used.

These measurements enable ultrafast, time-resolved access to transient heating, ionization, and recombination dynamics in hot dense plasmas, overcoming the limitations of previous time-integrated diagnostics. Although the total X-ray transmission is successfully derived from absorption imaging, reconstructing the spatial distribution of phase and attenuation maps remains challenging, owing to the limited imaging quality in the diffractive regime of this experiment[19]. With an improved setup—such as the newly commissioned Talbot X-ray imaging system at the XFEL facility[39], which incorporates an additional imaging lens and Talbot grating between the sample and detector—it is possible to simultaneously track the evolution of spatially resolved density and temperature at sub-micron resolution. This capability, combined with sensitive charge-state population diagnostics, provides a powerful tool for addressing key challenges in inertial confinement fusion (ICF)-relevant microphysics, including high-resolution imaging of early hydrodynamic instability growth[40], radiation asymmetry[41] and fuel mixing[42] of the burning core in future ICF implosion experiments with unprecedented precision. Consequently, this work is of broad interest to the high-energy-density and ICF communities, both as an experimental platform for accessing these theoretically challenging conditions and

as a benchmark for improving models of high-power laser-plasma interactions.

## Methods
### X-ray setup and parameters
The online calibrated HIgh REsolution hard X-ray single-shot (HIREX) spectrometer installed in the photon tunnel of the SASE2 undulator beamline was used to monitor the SASE spectrum[43] that ensured the central X-ray photon energy fixed at 8.2 keV during the entire beamtime. Fitting the measured X-ray spectrum with a Gaussian function gives the FWHM bandwidth of ~18 eV. The photon energy was cross-checked by the elastic X-ray scattering signal on the test Cu samples measured by the von Hámos X-ray spectrometer inside the interaction chamber[28]. The X-ray pulse energy was measured via an X-ray gas monitor (XGM) located in the photon tunnel and an intensity and position monitor (IPM) installed at the HED instrument[44]. The XFEL beam was focused onto the sample to a spot size with a full width at half maximum (FWHM) of ~10 μm, using the compound refractive lens (CRL) configuration comprising arms 3, 6, and 10 of CRL3, located in the HED optics hutch at a distance of ~962 m from the source and ~9 m to the target chamber center (interaction point)[17]. The XFEL spot size was characterized by the edge scan of the test samples via X-ray transmission using Zyla detector located at 6.31 m downstream from the target chamber center.

### ReLaX optical laser setup and parameters
The ReLaX optical laser is a 3 J, 30 fs, 10 Hz Ti:sapphire-based system, irradiating the Cu wires at normal incidence. The laser intensity profile measured by a ×20 APO PLAN microscope objective, showed that only a fraction of the total energy was contained in the central focal spot, deviating from a perfect Gaussian profile, as shown in Fig. 5. A statistical analysis of 618 ReLaX pulses yielded mean FWHM focal sizes of 3.9 μm (horizontal) and 3.6 μm (vertical), with relative standard deviations of 12% and 11%, respectively. The fraction of energy within the central focus was 0.44, with a relative standard deviation of 9%. Fluctuations in laser energy and pulse duration were negligible compared to other sources. Accounting for all these factors, the peak intensity was estimated to be $2.5 \times 10^{20}$ W/cm², with a relative standard fluctuation below 25%, corresponding to a range of $(1.875 - 3.125) \times 10^{20}$ W/cm². The temporal contrast of pulse intensity was measured by the offline diagnostics package consisting of single-shot temporal pulse diagnostics (SHG cross-correlator, a WIZZLER 800 for the spectrally resolved phase and intensity), a scanning third-order intensity autocorrelator (Sequoia HD) and a spatial phase sensor (SID4) in combination with a full beam adaptive deformable mirror, seen in our earlier work[19]. It is noted that the microscope objective was used only to generate a magnified image of the focal spot for focus-quality characterization during alignment. Because the objective was positioned downstream of the target chamber center, any dispersion it introduced would not affect the laser-matter interaction. Moreover, it was retracted during high-energy shots to prevent damage to the optics. Therefore, the microscope objective did not alter the laser pulse duration or the intensity on target.

### Synchronization between optical and X-ray lasers
The timing and synchronization between optical and X-ray laser were measured by the optical encoding spatial photon arrival monitor (PAM)[45]. The measurable timing window by the PAM is within 200 fs, which gives the upper limit of the uncertainty of XFEL probe time delay relative to the ReLaX laser presented in this work.

### Bremsstrahlung and self-emission background subtraction of RXES data
The approach of removing the Bremsstrahlung background subtraction of RXES data has three steps. First, we fit the off-axis background

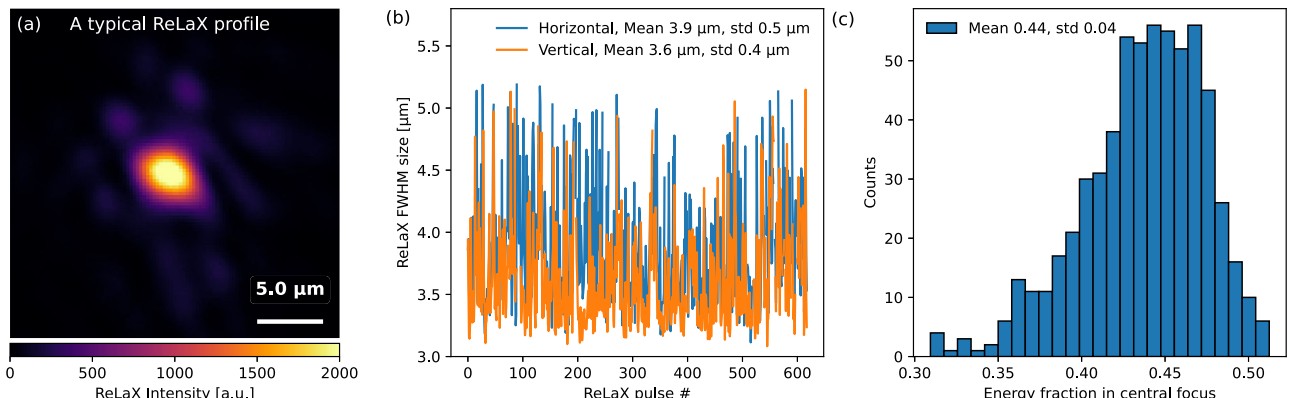

**Fig. 5 | Experimental measurements of ReLaX laser spatial profile. a** A typical example of the spatial intensity profile of ReLaX laser beam. **b, c** show the shot-to-shot statistical fluctuations of the FWHM size and the energy fraction within the central focus, respectively. The standard deviation is denoted by "std".

for each pixel coordinate along the dispersive direction. Then we use the fit and construct an on-axis background, including error. At last, the background is subtracted under the condition that the result is strictly non-negative (marginalizing over the error). The error on the spectrum is a combination of Poisson observation error, dark image root-mean-square (including multiple gain stages) and background correction error. Finally, to obtain the net resonant X-ray yield, the self-emission at ~8.2 keV is subtracted by fitting the Bremsstrahlung-subtracted emission yield to the $He_\alpha$ yield from shots with negative delays, which are analogous to the laser-only shots.

## Justification criteria of of high-quality shots

To minimize the initial uncertainties in the coupling conditions among the optical laser, XFEL beam, and wire target, we analyzed the statistical distributions of $K_\alpha$ and $He_\alpha$ emission yields, the XFEL transmission spatial profiles, and the total transmitted intensity for justification of high-quality ("good") shots. It is noted that here the $He_\alpha$ is defined as the $2p \to 1s$ transitions in $Cu^{26+}$ and $Cu^{27+}$ ranging from 8.28 to 8.42 keV. The $K_\alpha$ emission yield is found to be saturated, being limited by laser absorption and the resulting hot electron population as seen in Fig. 6. In contrast, the $He_\alpha$ yield serves as a more reliable indicator of plasma heating. Based on these statistics, high-quality ReLaX shots were selected by applying a threshold: $He_\alpha$ emission yields greater than $0.7 \times 10^{10}$ photons per steradian. These selected good shots were further filtered by requiring a total XFEL transmission below 0.9, suggesting good spatial overlap between the XFEL and the target. This overlap is also confirmed by examining the corresponding XFEL spatial profiles as seen in Fig. 7. The streaks observed in the good shots result from X-ray scattering at the edges of the wires. In contrast, when the XFEL transmission exceeds 0.9, the edge-scattering streaks are no longer apparent, indicating that the XFEL beam likely only partially hit the wires.

## PIC simulations

In this work, all the 2D3V PIC simulations are performed with PICLS code[36], employing the realistic laser and target parameters as used in the ReLaX laser pump-XFEL probe experiment. To quantify the impact of the laser parameters on the PIC simulations, the p-polarized laser pulse with the laser wavelength $\lambda_0 = 0.8\,\mu m$ and peak intensities ranging from $1.875 \times 10^{20}$ W/cm² to $5 \times 10^{20}$ W/cm², is modeled using Gaussian profiles in both space and time with full width at half maximum (FWHM) spot size $w_{FWHM} = 4\,\mu m$ and duration $\tau_{FWHM} = 30$ fs respectively. The laser irradiates the solid copper (Cu) wire target of 10 μm diameter at normal incidence. In order to take account the effect of leading edge of ReLaX laser pulse, the density profile of preplasma predicted by the MHD simulations is added to

the front surface of the Cu target. The mass density of Cu target is 8.96 g/cm³, corresponding to the number density of Cu ions being 48.27 $n_c$, where $n_c = 1.74 \times 10^{21}$ cm⁻³ is the plasma critical density. The cell size $\Delta x = \Delta y = 5.3$ nm and time step $\Delta t = 0.0178$ fs are set to resolve the plasma wavelength (21.4 nm) and frequency (0.071 fs) in fully ionized case that the free electron density reaches 1400 $n_c$. The simulation box consists of $N_x \times N_y = 7500 \times 5000$, corresponding to the real space size 40 μm × 27 μm. The axes $x$ and $y$ represent the directions of laser propagation and polarization respectively. The initial charge state of Cu ions is 1+ and the computational $Cu^{1+}$ ion number per cell is 5, corresponding to 145 electrons per cell for fully ionized case. The choice of fine cell size, time step and particle numbers ensures the elimination of numerical heating in our simulations. All the PIC simulations use the absorbing boundary and start with an initially cold neutral plasma. The collision ionization is treated either by the LTE based Thomas-Fermi pressure ionization model or NLTE based direct impact ionization model[8]. The field ionization and plasma heating is treated by the Landau-Lifshitz field ionization and the relativistic binary collisional operator respectively[36]. The large scale simulations using the realistic plasma density including both ionization and collision processes require huge computational resource to illuminate the numerical heating. Due to limited computational resources, the PIC simulations were performed up to 100 fs after the peak laser intensity reached the front surface of the target.

## MHD simulations

To assess the sensitivity of the predictive capability to the choice of EOS library, MHD simulations using the FLASH code[38] with the SESAME[46] and IONMIX[47] EOS are performed to simulate the preplasma formation driven by the rising slope of the ReLaX laser pulse ranging from −80 ps to −1 ps prior to the arrival of the peak ReLaX laser pulse, corresponding to the laser intensity rising from ~10¹² W/cm² to ~10¹⁶ W/cm². Figure 8 compares the spatial density profiles of preplasmas obtained using the SESAME and IONMIX EOS, respectively. Up to 1 ps before the arrival of the peak laser intensity, the preplasma density exhibits a short scale length near the initial solid target surface, extending into a much lower-density region with a longer scale length in both cases. While the scale lengths in the very low-density region are comparable, the near-surface scale length is slightly larger for IONMIX (~100 nm) than for SESAME (~60 nm). The predicted density profile is used as the input to the kinetic PIC simulations during the main pulse solid interactions. Then the PIC predicted spatial distribution of the plasma electron and ion temperatures at 100 fs are coupled to the FLASH MHD simulation to investigate the late time hydrodynamic compression and expansion driven by the shock waves from direct

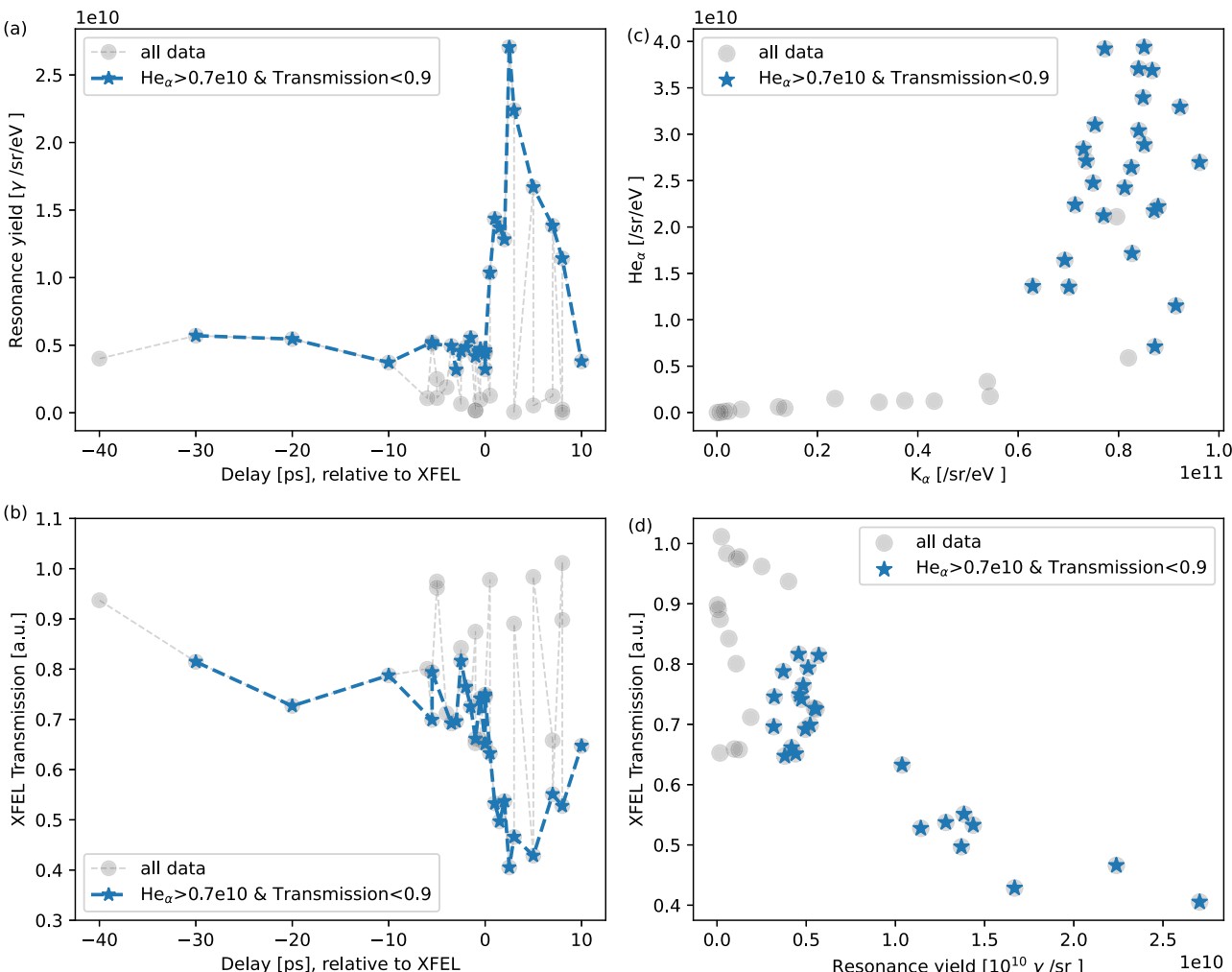

**Fig. 6 | Correlations of on-resonance, $K_\alpha$ and $He_\alpha$ emission yields, and XFEL transmission for all the hot shots.** Time-resolved resonant X-ray emission yield (**a**) and XFEL transmission (**b**). Correlation between $K_\alpha$ and $He_\alpha$ emission yields (**c**), and between resonant emission yield and XFEL transmission (**d**) for all the shots and selected good shots.

laser ablation and transient surface return current up to 150 ps after the laser matter interactions. The adaptive mesh size and time step are set to satisfy the Courant-Friedrichs-Lewy (CFL) conditions to ensure the convergence of the simulations. The combination of kinetic and MHD simulations enables us to explore the complex dynamics of electron transport during the entire relativistic laser solid wire interactions.

### Effect of peak laser intensity and preplasma profile

At present, predictive understanding of the complex dynamics in ultra-short, relativistic laser-solid interactions relies primarily on PIC simulations. However, for comparison with experimental data, PIC simulations have typically assumed a peak laser intensity corresponding to the full laser energy contained within an ideal Gaussian focus, together with a preplasma described by a simple exponential density profile[21]. As we discuss below, these simplified initial conditions can lead to significant discrepancies between simulated and measured plasma heating and ionization dynamics. Figure 9 shows how variations in the peak laser intensity and preplasma profile strongly affect the predictive capability of the PIC simulations. In the case of an MHD-simulated preplasma density profile with the SESAME EOS and a peak laser intensity of $5 \times 10^{20}$ W/cm², assuming that the full ReLaX energy is contained within ideal Gaussian profiles in both space and time, the

front surface temperature at the interaction point rises to ~12 keV and decreases to above 1 keV near the rear surface of the target. It results in the nearly full strip of L-shell electrons of Cu ions near the target surface and 5-7 µm in-depth ionization for the resonant charge state. This significantly overestimates plasma heating and ionization—most significantly at the surface, but also at depth, compared with the experimental observations. In contrast, using a more realistic peak intensity in the central focus, $1.875 \times 10^{20}$ W/cm², which corresponds to the lower-bound peak intensity accounting for the non-negligible fraction of energy distributed outside the central region, the surface temperature is reduced to ~5 keV and declines rapidly to ~1.5 keV within a depth of 1.5 µm. From the measured X-ray emission spectrum, no $Ly_\alpha$ lines are observed, indicating that the temperature in the solid region should remain below 1.5 keV based on SCFLY atomic simulations. Therefore, even with the corrected peak intensity, the simulation still overestimates the heating. By further adjusting the preplasma density profile based on the MHD simulations using the IONMIX EOS, the surface temperature is reduced to ~2.5 keV, falls to ~1.5 keV within a depth of 0.2 µm, and decreases below 1 keV throughout the target bulk. Correspondingly, the predicted resonant charge-state of interest is populated near the surface, consistent with the resonant emission and absorption imaging of highly charged Cu ions observed in the experiment, which reveals localized surface heating

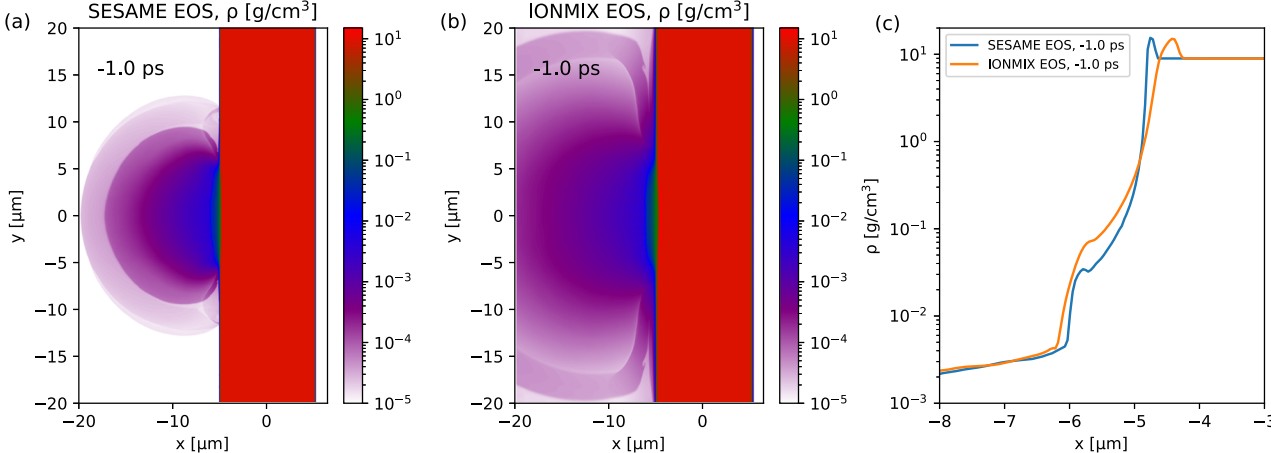

**Fig. 7 | An overview of the spatial distribution of X-ray transmission imaging profiles.** Its correlation with Kα, Heα, and resonant emission yields (denoted as "Res"), the X-ray pulse energy measured by the IPM (denoted as "IPM"), as well as total XFEL transmission for all the shots, is shown alongside the imaging data. The good shots discussed in the main text are marked with black titles (satisfying the conditions of He$_\alpha$ emission yield exceeding $0.7 \times 10^{10}$ photons per steradian and XFEL transmission below 0.9), while the others are indicated by gray titles. The spatial scale bar corresponds to 10 μm for all images.

**Fig. 8 | 2D MHD simulation results for preplasma conditions used as input for subsequent PIC simulations.** Preplasma density profiles obtained using the SESAME **a** and IONMIX **b** EOS libraries up to 1 ps before the arrival of the peak laser intensity. **c** Comparison of longitudinal lineouts along the target center.

confined to the laser spot size. Even when the peak laser intensity is increased to the measured upper bound of $3.125 \times 10^{20}$ W/cm² within the standard deviation, the simulated resonant charge state remains predominantly confined near the surface, extending only to ~2 μm into the target, and still agrees with the experimentally inferred localization. Thus, the systematic simulations and adjustments clearly show that the realistic peak laser intensity and the preplasma density profile constitute the most critical initial conditions for reliably predicting the plasma heating and ionization dynamics observed experimentally.

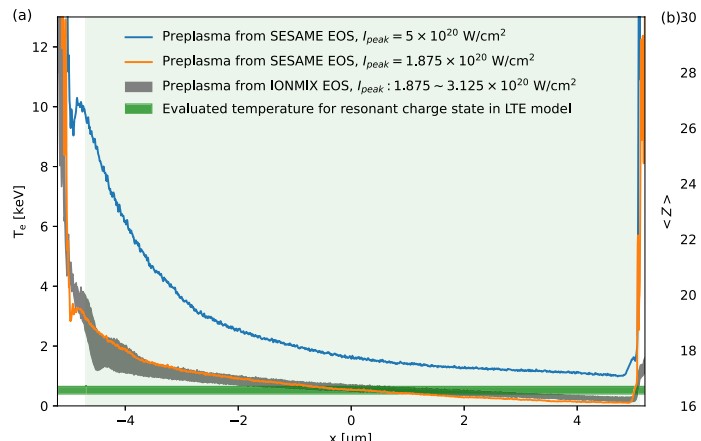
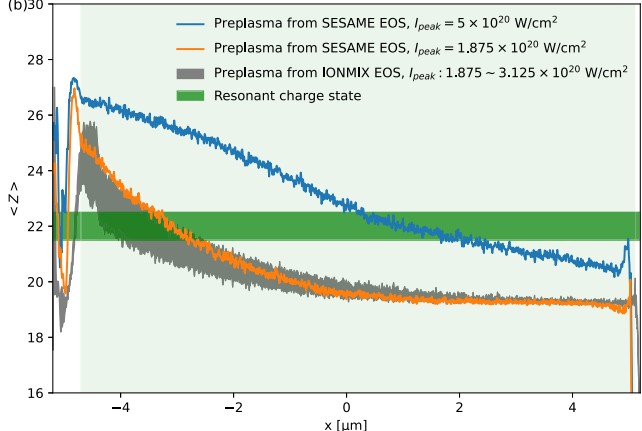

**Fig. 9 | 2D PIC simulations of the effect of peak laser intensity and preplasma profile on the heating and ionization dynamics.** The simulations are based on a flat geometry, corresponding to a side view of the solid Cu wire target, extracted 100 fs after the main pulse arrival. Comparisons of longitudinal lineouts of **a** the plasma temperature and **b** the average ion charge state along the target center for different interaction conditions, with the initial solid target region indicated by the light green background.

## Data availability

The data recorded for the experiment at the European XFEL are available at EuXFEL data repository, HED 3129, after the expiration of the embargo period[48] or from the corresponding author, L.H., upon request. The simulation data used to generate Figs. 1, 4 and 8–9 are available at the Rossendorf data repository[49].

## Code availability

The MHD code FLASH 4.8 used in this work was developed in part by the DOE NNSA- and DOE Office of Science-supported Flash Center for Computational Science at the University of Chicago and the University of Rochester, and it is publicly available. The PIC code PICLS and analysis scripts used in this study are available upon reasonable request.

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

## Acknowledgements

We acknowledge European XFEL in Schenefeld, Germany, for the provision of X-ray free-electron laser beamtime at HED SASE2 under proposal number 3129. The authors thank the HiBEF user consortium and their staff for their user support and the equipment provided to make this experiment possible. The authors would also like to thank the HZDR and DESY HPC group for their assistance on the high-performance computational clusters. L.H. gratefully acknowledges Prof. O. Rosmej, Dr J. F. Ong, Dr A. Hirsch-Passicos, and C. Qu for valuable discussions. T. Engler acknowledges the funding of Grant No. HIDSS-0002 DASHH (Data Science in Hamburg - Helmholtz Graduate School for the Structure of Matter).

## Author contributions

L.H. led the experiment. L.H., M.M., M.S, T.R.P., T.K., A.L.G., H-P.S., T.T., U.Z. and T.E.C conceived the experimental setup. L.H., M.M., M.S., T.R.P., X.P., L.Y., T.E., C.B., E.B., A.L.G., S.G., M.H., H.H., M.K., M.Masuri, M.N., M.O., Ö.Ö., A.P., L.R., M.R., H-P.S., J-P.S., M.T., T.T., and T.E.C performed the experiment. L.H., M.M, M.S., O.S.H., T.R.P., X.P., L.Y., J.H., T.E., Y.C., T.K., A.L.G., T.T., U.Z., and T.E.C. analyzed the data. L.H. wrote the original manuscript draft. All authors, including C.G., J.M-N., I.P., U.S., J.V., and K.Z., discussed the results and revised the manuscript.

## Funding

## Competing interests

The authors declare no competing interests.

## Additional information

Lingen Huang [1] ✉, Mikhail Mishchenko [2,3], Michal Šmíd [1], Oliver S. Humphries [2], Thomas R. Preston [2], Xiayun Pan [1,4], Long Yang [1,4], Johannes Hagemann [5], Thea Engler [5], Yangzhe Cui[1], Thomas Kluge [1], Carsten Baehtz [1], Erik Brambrink[2], Alejandro Laso Garcia [1], Sebastian Göde[2], Christian Gutt [6], Mohamed Hassan[1], Hauke Höppner [1], Michaela Kozlova[1,7], Josefine Metzkes-Ng [1], Masruri Masruri[1], Motoaki Nakatsutsumi [2], Masato Ota[8], Özgül Öztürk [6], Alexander Pelka[1], Irene Prencipe[1], Lisa Randolph [2], Martin Rehwald [1], Hans-Peter Schlenvoigt [1], Ulrich Schramm [1,4], Jan-Patrick Schwinkendorf[1], Monika Toncian[1], Toma Toncian [1], Jan Vorberger [1], Karl Zeil [1], Ulf Zastrau [2] & Thomas E. Cowan [1,4]

¹Helmholtz-Zentrum Dresden-Rossendorf, Dresden, Germany. ²European XFEL, Schenefeld, Germany. ³Technische Universität Bergakademie Freiberg, Freiberg, Germany. ⁴Technische Universität Dresden, Dresden, Germany. ⁵Center for X-ray and Nano Science CXNS, Deutsches Elektronen-Synchrotron DESY, Hamburg, Germany. ⁶Universität Siegen, Siegen, Germany. ⁷The Extreme Light Infrastructure ERIC, ELI Beamlines Facility, Dolní Br^ežany, Czech Republic. ⁸National Institute for Fusion Sciences, Toki, Japan. ✉e-mail: lingen.huang@hzdr.de

