## [Peer Review file · Nature Communications]

Probing ultrafast heating and ionization dynamics in solid density plasmas with time-resolved resonant X-ray absorption and emission

Corresponding Author: Dr Lingen Huang

Version 0:

Reviewer comments:

Reviewer #1

(Remarks to the Author)

This manuscript reports an experimental investigation of the spatiotemporal evolution of the temperature and ionization state in intense laser-irradiated copper wire targets. These quantities underlie key plasma properties — temperature, density, resistivity, and opacity — that are central to both fundamental high-energy-density physics and a wide range of applications.

The topic is timely and relevant. Laser-driven plasmas remain exceptionally challenging to characterize because they feature rapidly evolving, strongly nonequilibrium conditions, steep gradients, and relativistic electron populations. Extracting reliable, time-dependent information on atomic excitation, ionization, and relaxation in such regimes is particularly difficult and remains a bottleneck for validating theory and simulation.

The authors present simultaneous time-resolved resonant X-ray emission spectroscopy and X-ray absorption imaging of Cu wires, employing a coherent XFEL probe tuned to the $1s \rightarrow 2p$ resonance of highly ionized copper ions. The methodology is appropriate and represents an ambitious coupling of laser-driven plasma generation with XFEL-based pump-probe diagnostics. By scanning the delay between the infrared drive laser and the XFEL (8.2 keV), the authors track the resonant emission from Cu^{22+} , observing a ~ 10 ps rise-and-fall behavior, peaking near 2.5 ps, in inverse correlation with the X-ray transmission. From these data, they infer the temporal evolution of the population in this charge state, and relate it to the evolution of the plasma temperature and opacity.

The observation of off-resonant line emission arising from Auger and collisional relaxation is noteworthy. Although similar signatures have been reported in XFEL-only experiments, their appearance in a laser-driven, highly inhomogeneous plasma with a non-thermal electron distribution (including a relativistic tail) is challenging to interpret. The comparable intensities of non-resonant features on both sides of the XFEL photon energy suggest a balance between ionization and recombination processes.

The modeling framework is extensive: MHD simulations to characterize the pre-plasma; PIC simulations with collisional-radiative atomic physics to describe the laser interaction, ionization dynamics, relativistic electron transport, and isochoric heating; and subsequent MHD modeling of the hydrodynamic evolution. This multi-stage approach is ambitious, but also introduces significant sources of uncertainty. The PIC simulations that incorporate nonequilibrium collisional processes produce ionization-state predictions closer to experiment than LTE-based models, but still systematically overestimate plasma heating—most significantly at the surface, but also at depth. This mismatch points to limitations either in the assumed pre-plasma conditions, the treatment of non-thermal electrons, or the collisional-radiative rates under warm-dense-matter conditions. The manuscript would benefit from a more explicit discussion of these uncertainties, including their potential impact on the inferred charge-state distributions.

The need for improved characterization of the laser temporal contrast is clear, as it would enable more realistic modeling of pre-plasma density gradients and thereby reduce uncertainty in the PIC-MHD chain. Similarly, advances in non-LTE collisional-radiative modeling in the warm dense matter regime remain necessary for truly quantitative comparison.

Despite these limitations, the study presents a meaningful advance relative to the existing literature. Its key strengths

include:

- i) The use of XFEL radiation simultaneously as a pump and probe, achieving sub-picosecond temporal and sub-micrometer spatial resolution;
- ii) The generation and diagnosis of warm dense matter under relativistic-laser conditions coupled to an XFEL environment;
- iii) The use of thin cylindrical targets, which effectively mitigate uncertainties from XFEL spatial and temporal jitter.

Finally, the longer (~ 10 ps) and stronger resonant emission observed in Cu wires, compared to previously studied Cu foils, is a valuable result. It highlights enhanced laser-energy coupling due to reduced target mass and mitigated hot-electron divergence, and further validates the experimental design.

In summary, while certain aspects of the modeling and interpretation require refinement and clearer quantitative justification, the experimental approach is innovative and the results are both original and scientifically relevant. Overall, the manuscript constitutes a significant and publishable contribution to the field of high-energy-density physics, and it is suitable for publication after minor revision.

Reviewer #2

(Remarks to the Author)

This study reports on time-resolved plasma temperature diagnostics using an XFEL. The use of resonant absorption measurements is highly novel, and the selection of high-quality data yields clean and reliable results. The manuscript is well written and easy to follow.

However, the experimental results themselves are quite basic and largely as expected, lacking impact or surprise. Therefore, I do not recommend this manuscript for publication in Nature Communications.

My specific comments are as follows:

1. The physical implications of these experimental results for the field should be carefully discussed. In particular, if the simulations do not reproduce the measurements, the reasons for the discrepancies should be quantitatively analyzed.
2. The authors state that the 2D-PIC results do not agree with the experiment. However, they should present the spatially weighted X-ray spectrum or transmission calculated from the simulation and compare it directly with the experimental data.
3. The correlation between the resonant X-ray emission yield and the XFEL transmission shown in Fig. 2(c) requires a clear physical discussion, ideally supported by comparison with theoretical or simulation results.
4. The distinction between the present diagnostic method and those in previous studies is weak. The authors should quantitatively describe the advantages of their approach compared with existing techniques.
5. The manuscript mentions that a microscope objective was used in the final optical system. Has it been verified that the pulse duration was not stretched due to geometric path length differences or group-velocity dispersion (GVD)? Such stretching could significantly alter the laser intensity.
6. Figure 2 in the Methods section is not referenced anywhere in the main text. In addition, the spatial scale should be indicated on the image.

Version 1:

Reviewer comments:

Reviewer #1

(Remarks to the Author)

This referee is satisfied with the improvements in the manuscript, which address the questions/criticism from the reviewers. The more realistic account of the laser intensity profile in the simulations was particularly appreciated.

Reviewer #2

(Remarks to the Author)

The authors have responded appropriately and thoroughly to the reviewer comments.

In the revised manuscript, the experimental results are well compared with the simulation results, representing a significant improvement over the previous version.

I therefore recommend this manuscript for publication.

Probing ultrafast heating and ionization dynamics
in solid density plasmas with time-resolved resonant X-ray absorption and emission

Response to Reviewer #1

Reviewer #1 (Remarks to the Author):

This manuscript reports an experimental investigation of the spatiotemporal evolution of the temperature and ionization state in intense laser-irradiated copper wire targets. These quantities underlie key plasma properties — temperature, density, resistivity, and opacity — that are central to both fundamental high-energy-density physics and a wide range of applications.

The topic is timely and relevant. Laser-driven plasmas remain exceptionally challenging to characterize because they feature rapidly evolving, strongly nonequilibrium conditions, steep gradients, and relativistic electron populations. Extracting reliable, time-dependent information on atomic excitation, ionization, and relaxation in such regimes is particularly difficult and remains a bottleneck for validating theory and simulation.

The authors present simultaneous time-resolved resonant X-ray emission spectroscopy and X-ray absorption imaging of Cu wires, employing a coherent XFEL probe tuned to the $1s \rightarrow 2p$ resonance of highly ionized copper ions. The methodology is appropriate and represents an ambitious coupling of laser-driven plasma generation with XFEL-based pump-probe diagnostics. By scanning the delay between the infrared drive laser and the XFEL (8.2 keV), the authors track the resonant emission from Cu^{22+} , observing a ~ 10 ps rise-and-fall behavior, peaking near 2.5 ps, in inverse correlation with the X-ray transmission. From these data, they infer the temporal evolution of the population in this charge state, and relate it to the evolution of the plasma temperature and opacity.

The observation of off-resonant line emission arising from Auger and collisional relaxation is noteworthy. Although similar signatures have been reported in XFEL-only experiments, their appearance in a laser-driven, highly inhomogeneous plasma with a non-thermal electron distribution (including a relativistic tail) is challenging to interpret. The comparable intensities of non-resonant features on both sides of the XFEL photon energy suggest a balance between ionization and recombination processes.

The modeling framework is extensive: MHD simulations to characterize the pre-plasma; PIC simulations with collisional-radiative atomic physics to describe the laser interaction, ionization dynamics, relativistic electron transport, and isochoric heating; and subsequent MHD modeling of the hydrodynamic evolution. This multi-stage approach is ambitious, but also introduces significant

sources of uncertainty. The PIC simulations that incorporate nonequilibrium collisional processes produce ionization-state predictions closer to experiment than LTE-based models, but still systematically overestimate plasma heating—most significantly at the surface, but also at depth. This mismatch points to limitations either in the assumed pre-plasma conditions, the treatment of non-thermal electrons, or the collisional–radiative rates under warm-dense-matter conditions. The manuscript would benefit from a more explicit discussion of these uncertainties, including their potential impact on the inferred charge-state distributions.

The need for improved characterization of the laser temporal contrast is clear, as it would enable more realistic modeling of pre-plasma density gradients and thereby reduce uncertainty in the PIC–MHD chain. Similarly, advances in non-LTE collisional–radiative modeling in the warm dense matter regime remain necessary for truly quantitative comparison.

Despite these limitations, the study presents a meaningful advance relative to the existing literature.

Its key strengths include:

- i) The use of XFEL radiation simultaneously as a pump and probe, achieving sub-picosecond temporal and sub-micrometer spatial resolution;
- ii) The generation and diagnosis of warm dense matter under relativistic-laser conditions coupled to an XFEL environment;
- iii) The use of thin cylindrical targets, which effectively mitigate uncertainties from XFEL spatial and temporal jitter.

Finally, the longer (~ 10 ps) and stronger resonant emission observed in Cu wires, compared to previously studied Cu foils, is a valuable result. It highlights enhanced laser-energy coupling due to reduced target mass and mitigated hot-electron divergence, and further validates the experimental design.

In summary, while certain aspects of the modeling and interpretation require refinement and clearer quantitative justification, the experimental approach is innovative and the results are both original and scientifically relevant. Overall, the manuscript constitutes a significant and publishable contribution to the field of high-energy-density physics, and it is suitable for publication after minor revision.

Our response:

We sincerely acknowledge the referee for their careful review of our manuscript and for the positive feedback that our experimental approach is innovative, the results are both original and scientifically

relevant, and the work constitutes a significant and publishable contribution to the field of high-energy-density physics.

We also thank the referee for their constructive comments, noting that certain aspects of the modeling and interpretation require refinement and clearer quantitative justification.

To minimize uncertainties in the multi-stage modeling framework, we examined the experimentally measured spatial profiles of ReLaX laser intensity and re-performed MHD simulations with different equation of state (EOS) libraries to characterize the preplasma, which served as input for the subsequent hybrid PIC-MHD simulation chain. **We found that the multi-scale simulations reveal extreme sensitivity of basic plasma parameters with widely used models, such as temperature and ionization depth, which are able to be constrained by incorporating a detailed accounting of laser spatial profiles, and preplasma conditions, and NLTE collisional processes.**

Firstly, we examined the experimentally measured spatial profiles of ReLaX laser intensity and found that only a fraction of the laser energy is contained in the measured central focal spot, as shown in Fig.1. In the original manuscript, the simulations used a peak intensity of $5e20W/cm^2$, assuming a Gaussian ReLaX laser profile in the transverse plane and in time, with a $4\ \mu m$ FWHM spot size and a 30 fs FWHM pulse duration, respectively. However, the actual laser focus deviates from a perfect Gaussian profile, with a non-negligible fraction of the energy distributed outside the central region, as shown in Fig. 1. A statistical analysis of 618 ReLaX pulses shows mean FWHM focal sizes of $3.9\ \mu m$ (horizontal) and $3.6\ \mu m$ (vertical), with relative standard deviations of 12% and 11%, respectively. The fraction of energy contained within the central focus is 0.44, with a relative standard deviation of 9%. Fluctuations in laser energy and pulse duration are very low and thus negligible compared to other sources. Combining all of these factors results in a reduction of the peak intensity by a factor of 2 relative to the value used in the original manuscript, yielding $2.5e20W/cm^2$, with a relative standard fluctuation of less than 25%.

Figure 1: (a) A typical example of the spatial intensity profile of ReLaX laser beam. Panels (b) and (c) show the shot-to-shot statistical fluctuations of the FWHM size and the energy fraction within the central focus, respectively.

Secondly, in addition to the MHD simulations using the SESAME equation of state (EOS) [SESAME1994] presented in the original manuscript, we re-ran the MHD simulations using the IONMIX EOS [IONMIX1989] to characterize the preplasma density profile generated by the intrinsic laser rising edge or prepulse, and to assess the sensitivity of the results to the choice of EOS. Figure 2 compares the spatial density profiles of preplasmas obtained using the SESAME and IONMIX EOS. Up to 1 ps before the arrival of the peak laser intensity, the preplasma density exhibits a short scale length near the initial solid target surface, extending into a much lower-density region with a longer scale length in both cases. While the scale lengths in the very low-density region are comparable, the near-surface scale length is slightly larger for IONMIX (~100 nm) than for SESAME (~60 nm), which plays a crucial role in laser absorption at the arrival of the peak laser intensity. Using the nanometer-sensitive small-angle X-ray scattering (SAXS) diagnostic, our recent work demonstrates that the MHD-predicted preplasma density profile from IONMIX agrees better with the experimental SAXS measurements [Huang2026], providing strong validation for the choice of IONMIX EOS in the simulations.

Figure 2: Comparison of the spatial preplasma density profiles obtained using the SESAME and IONMIX EOS up to 1 ps before the arrival of the peak laser intensity.

Based on a detailed assessment of the PIC simulation initial conditions, we re-performed hybrid MHD–PIC simulations using the lower-bound peak laser intensity in the central focus ($2.5e20W/cm^2 * 0.75 = 1.875e20W/cm^2$) and varied preplasma density profiles, comparing the

results to those in the original manuscript in the side view of a wire target (flat geometry). As shown in Fig. 3, the simulation using a peak intensity of $5 \times 10^{20} \text{ W/cm}^2$ overestimates the plasma heating and ionization—most significantly at the surface, but also at depth. In contrast, when using the more realistic peak intensity in the central focus, $1.875 \times 10^{20} \text{ W/cm}^2$, the surface temperature is reduced from $\sim 12 \text{ keV}$ to $\sim 5 \text{ keV}$ and falls to $\sim 1.5 \text{ keV}$ within a depth of $1.5 \mu\text{m}$. From the measured X-ray emission spectrum, no Ly_α lines are observed, indicating that the temperature in the solid region should remain below 1.5 keV based on SCFLY atomic simulations. Therefore, even with the corrected peak intensity, the simulation still overestimates the heating. By further adjusting the preplasma density profile using the MHD prediction with the IONMIX EOS (instead of the SESAME EOS), the surface temperature is reduced to $\sim 2.5 \text{ keV}$, drops to $\sim 1.5 \text{ keV}$ within a depth of $0.2 \mu\text{m}$, and then decreases below 1 keV in the target bulk. Correspondingly, the resonant charge-state of interest is populated near the surface, consistent with the measured resonant X-ray emission yield and the XFEL absorption which reveal the small scale length of the resonant feature and localized surface heating confined to the laser spot size. Even when the peak laser intensity is increased to the measured upper bound of $3.125 \times 10^{20} \text{ W/cm}^2$ within the standard deviation, the simulated resonant charge state remains predominantly confined near the surface, extending only to $\sim 2 \mu\text{m}$ into the target, and remains in satisfactory agreement with the measurement.

In conclusion, we examined the experimentally measured spatial profiles of the ReLaX laser intensity and re-performed the MHD simulations using different EOS libraries to obtain more accurate preplasma conditions for the hybrid PIC–MHD simulation chain. This systematic adjustment shows that the realistic peak laser intensity and the preplasma density profile constitute the most critical initial conditions for reliably predicting the plasma heating and ionization dynamics observed experimentally.

Figure 3: Longitudinal lineouts of (a) the plasma temperature and (b) the average ion charge state for the side view of a wire target (flat geometry), with the initial solid target region indicated by the light green background.

In the revised manuscript, we have added Fig. 1 and Fig. 2 to the subsections “*ReLaX optical laser setup and parameters*” and “*MHD simulations*” within the METHODS section. In addition, we have introduced a new subsection, “*Effect of initial laser and target conditions on the plasma heating and ionization,*” which includes Fig. 3. This new subsection highlights how the choice of initial conditions critically influences the simulation results and, consequently, the quantitative comparison with the experimental data. Furthermore, the hybrid PIC–MHD results shown in Fig. 4 of the original manuscript have been updated with new simulations incorporating the corrected peak laser intensity and preplasma density profile, showing good consistency with the experimental measurements. The updated simulation results are also presented below.

In conclusion, we found that the multi-scale simulations reveal extreme sensitivity of basic plasma parameters with widely used models, such as temperature and ionization depth, which are able to be constrained by incorporating a detailed accounting of laser spatial profiles, and preplasma conditions, and NLTE collisional processes. Correspondingly, we have substantially revised the discussion of the simulation results and their comparison with the experimental data in the main text. In addition, we have included further simulations that systematically scan the initial interaction conditions, including the laser peak intensity and preplasma density profiles, in the METHODS section of the Appendix. All changes in the manuscript are highlighted in red.

Figure 4: Updated hybrid PIC-MHD simulations with corrected peak laser intensity and the preplasma density profile. Spatial distribution of electron temperature and ion charge state obtained from 2D PIC simulations in the case of top view of a Cu wire with $10\ \mu\text{m}$ thickness, using (a) the LTE based Thomas-Fermi and (b) the NLTE based direct impact ionization models, both at 100 fs after peak laser intensity irradiation. Corresponding 2D PIC and MHD simulations for the side view of the Cu wire using the NLTE ionization model are shown at (c) 100 fs and (d) 2.5 ps, respectively. Longitudinal lineouts of the plasma temperature (e) and average ion charge state (f) of the corresponding 2D simulation maps, extracted from the initial target region indicated by the light green area. The green regions indicate the probed resonant charge state and its corresponding evaluated plasma temperature.

[SESAME1994] J. Johnson, “The sesame database,” Tech. Rep. (1994)

[IONMIX1989] J.J. Macfarlane, Computer Physics Communications 56, 259 (1989)

[Huang2026] L. Huang, et al., Matter and Radiation at Extremes 11, 017201 (2026)

Response to Reviewer #2

Reviewer #2 (Remarks to the Author):

This study reports on time-resolved plasma temperature diagnostics using an XFEL. The use of resonant absorption measurements is highly novel, and the selection of high-quality data yields clean and reliable results. The manuscript is well written and easy to follow.

However, the experimental results themselves are quite basic and largely as expected, lacking impact or surprise. Therefore, I do not recommend this manuscript for publication in Nature Communications.

Our response:

High-quality, time resolved data on charge state populations in laser produced plasmas are valuable input to benchmark simulation data, and we thank the reviewer for identifying the work that has gone into producing these novel and high-quality data, yielding clean and reliable results. We agree that the essential result of observing resonance peaking with time was expected, as its measurement was the primary goal of the experiment. However, as demonstrated by the complex modeling presented in the revised manuscript, making precise quantitative predictions of this behavior is extremely challenging. Therefore, though qualitatively expected, the novel and high-quality data is critically important for precise understanding of the physics of extreme states of matter. The detailed simulation framework needed to approximately reproduce experimental data, and their sensitivity to the ionization models and input parameters, such as the realistic peak laser intensity and preplasma profile presented in the revised manuscript – highlight that these experimental results cannot be viewed as “expected, lacking impact or surprise”. As noticed by Reviewer #1 *“Extracting reliable, time-dependent information on atomic excitation, ionization, and relaxation in such regimes is particularly difficult and remains a bottleneck for validating theory and simulation,”* and we therefore believe our revised manuscript (all changes in the manuscript are highlighted in red), together with the newly performed extensive and detailed comparisons against state-of-the-art simulation capabilities directly refutes this claim – highlighting deficiencies in current modelling capabilities in extreme conditions, and how overreliance on these can result in order of magnitude errors in basic quantities such as temperature, as detailed in our response below.

My specific comments are as follows:

1. The physical implications of these experimental results for the field should be carefully discussed. In particular, if the simulations do not reproduce the measurements, the reasons for the discrepancies should be quantitatively analyzed.

Our response 1:

We fully agree with it.

We have examined the experimentally measured spatial profiles of the ReLaX laser intensity and re-performed the MHD simulations using different EOS libraries to characterize the preplasma, which further served as input for the subsequent hybrid PIC-MHD simulation chain. With additional extensive PIC and MHD simulations, we found that the realistic peak laser intensity based on measured particular laser spatial profiles and appropriate preplasma density profiles are the most critical initial conditions for the simulations to correctly predict the plasma heating and ionization dynamics observed experimentally. Further details and updated simulation results are included in our response to Reviewer #1, who raised similar comments.

2. The authors state that the 2D-PIC results do not agree with the experiment. However, they should present the spatially weighted X-ray spectrum or transmission calculated from the simulation and compare it directly with the experimental data.

Our response 2:

The reviewer is correct that a spectral postprocessing is a logical next step in understanding the situation, which requires to connect the 2D-PIC and MHD simulation results with collisional radiative atomic code. However, we note that ultrafast relativistic laser-driven plasmas remain exceptionally challenging to characterize due to their rapidly evolving, strongly nonequilibrium conditions, steep gradients, and relativistic electron populations. To illustrate this, we performed spectral postprocessing with atomic code FLYCHK using plasma parameters from 2D-PIC and MHD simulations as initial conditions, and found that NLTE opacities and non-thermal electrons are likely essential for reproducing the measured X-ray spectra and transmission.

To reproduce experimentally measured X-ray spectrum and transmission shown in Fig. 2 (b) of the manuscript, each cell of the 2D-PIC or MHD simulation was assigned a set of three spectra calculated by the FLYCHK code based on the cell's plasma density and temperature: an absorption spectrum, and an emission without any driver and emission with XFEL as a driver. The absorption

spectrum was used to evaluate a 2D absorption profile of the 8.2 keV peak, and the difference between the driven and undriven emission to show the increase of emission peak, as observed in the experiment. An example of simulation results for time delay at $t=4.1$ ps is shown in Fig.1 below.

Figure 1: 2D spatial distributions of mass density and electron temperature extracted from PIC–MHD simulations, along with simulated total X-ray self-emission, X-ray emission and absorption at 8.2 keV with an XFEL driver, and the difference between the driven and undriven X-ray emission at 8.2 keV.

We then integrated these spectra over the 2D spatial domain to reproduce the spatially weighted, experimentally measured X-ray spectrum or transmission, as shown in Fig. 2. It can be seen that in the standard atomic simulation, based on the MHD-predicted 2D temperature distribution (black lines), the peak emission and minimum absorption occur at approximately 11–15 ps, in contrast to the experimentally observed 2–3 ps. A strong difference is observed when postprocessing is performed on the MHD simulation using temperatures interpolated from charge state in the PIC code (dotted lines). In this case, the extremes of the observables (peak emission and absorption) disappear entirely because the temperatures are too low. As an indicator of the underlying mechanisms, a set of postprocessing tests using MHD temperatures scaled by different multipliers shows that the emission peak shifts to earlier times, reaching 3 ps as observed from the experiment when the temperature is reduced to about 70%. That is obviously not a realistic plasma simulation, but indicates the mechanism of probing given atomic transition.

Although the postprocessing simulation is preliminary, it highlights the critical role of transitioning from PIC to MHD results, as well as the non-thermal nature of the phenomena. Including non-thermal, non-equilibrium processes in predictive atomic modeling to accurately simulate time-dependent X-ray emission or transmission is particularly challenging, due both to the different thermalization timescales of electrons and ions and to the large number of atomic configurations required to describe highly ionized plasmas. Since improving and validating the atomic code is not the main focus of this work, including all associated details, caveats, and conclusions from the spectral postprocessing would overly scatter the current article, and is therefore better suited for a separate publication. Meanwhile, publishing these high-quality experimental data in the spirit of open science will allow other groups to benchmark and validate alternative atomic codes.

Figure 2: Spatially weighted resonant X-ray emission and XFEL transmission at 8.2 keV for different treatments of the electron temperature, as predicted by simulations using the atomic code FLYCHK.

3. The correlation between the resonant X-ray emission yield and the XFEL transmission shown in Fig. 2(c) requires a clear physical discussion, ideally supported by comparison with theoretical or simulation results.

Our response 3:

The observed correlation between the resonant X-ray emission yield and the XFEL absorption in

Fig. 2(c) indicates the small-scale length of the resonant features, as otherwise absorption becomes saturated, while backward resonant emission yield is dominated by the first absorption length thickness. In this way having both resonant backscatter (sensitive to the x-ray absorption depth) and transmission (bulk) are complementary diagnostics yielding additional information from their combination. For the resonant charge state corresponding to peak absorption at the XFEL photon energy, the effective absorption length is on the order of a micrometer, $O(\mu\text{m})$. This implies that, for linear proportionality, the projected thickness must be smaller than this value, further indicating that the ionization is highly localized near the front surface, rather than producing a uniform charge state. This contrasts with the early simulations in the original manuscript, which predicted ionization throughout the entire wire, but is consistent with the new simulation results showing localized ionization near the front surface when accurate initial interaction conditions are used. The small-scale length of the resonant feature is further supported by data obtained from $2\ \mu\text{m}$ CH-coated Cu wires (not shown in the current manuscript), for which no obvious resonant emission is observed. It is noted that although the features are smaller than the XFEL focal size, variations in direct beam overlap arising from beam jitter introduce only minor fluctuations in the total absorption signal. Forward X-ray propagation simulations show that the absorption variation is less than 0.1 assuming $\pm 5\ \mu\text{m}$ jitter, which is consistent with the experimental data shown in Fig. 2(b) at negative delays. Consequently, changes in the resonant absorption remain clearly distinguishable when the effects of beam jitter are taken into account.

Thus, while resonant emission serves as an ionization diagnostic and is extensible to thicker or layered targets containing a suitable resonance due to its spectral sensitivity, the observed correlation in resonant absorption can be used here to infer an approximate scale length even without direct imaging.

We have added the above discussion for Fig. 2(c) in the revised manuscript.

4. The distinction between the present diagnostic method and those in previous studies is weak. The authors should quantitatively describe the advantages of their approach compared with existing techniques.

Our response 4:

As per Reviewer #1 “...the study presents a meaningful advance relative to the existing literature. Its key strengths include: i) The use of XFEL radiation simultaneously as a pump and probe,

achieving sub-picosecond temporal and sub-micrometer spatial resolution ...”, and the present diagnostic provides quantitative information not accessible with previously established X-ray or optical techniques. Earlier ultrafast studies relied on either broadband XAS/XANES, K_{α} spectroscopy, or integrated bremsstrahlung emission, which lack charge-state specificity, are insensitive to narrow resonances, or average over the full charge-state distribution. In contrast, by tuning the XFEL to the Cu^{22+} $1s \rightarrow 2p$ resonance at 8.2 keV and simultaneously measuring the monochromatic attenuation of the probe and the resonantly driven emission, we isolate the contribution of a single charge state with sub-picosecond time resolution. This enables quantitative retrieval of both the transient history of Cu^{22+} population and the resonant opacity at solid density. Moreover, the tunability of the XFEL photon energy enables charge-sensitive diagnostics capable of measuring the spatiotemporal evolution of heating and ionization dynamics for any target of interest, which is clearly a significant advantage.

In our experiment, the technique achieves a high precision in transmission and emission measurements simultaneously and reveals that the hot-plasma absorption at the resonance exceeds the cold-Cu opacity by more than a factor of two. No previous diagnostic has provided simultaneous, time-resolved access to both resonant absorption and charge-state-selective emission in a highly transient, solid-density plasma. As also mentioned in the response 3, while resonant emission can serve as an ionization diagnostic—which can be extended to thicker or layered targets containing a suitable resonance due to its spectral sensitivity—the correlated resonant absorption can provide an estimate of the scale length even without imaging. Such a unique combination of diagnostic capabilities, with unprecedented simultaneous spatiotemporal resolution, allows for a comprehensive investigation of multi-scale physics by directly comparing experimental results with multi-scale simulations.

We have highlighted the distinct capabilities and benefits of the combined diagnostics in comparison with current techniques in the revised manuscript.

5. The manuscript mentions that a microscope objective was used in the final optical system. Has it been verified that the pulse duration was not stretched due to geometric path length differences or group-velocity dispersion (GVD)? Such stretching could significantly alter the laser intensity.

Our response 5:

The microscope objective mentioned in the manuscript was used only to generate a magnified image of the focal spot for focus-quality characterization in an alignment mode. Because the objective was positioned downstream of the target chamber center (TCC), any dispersion it introduced would not affect the laser-matter interaction. Moreover, it was retracted during high-energy shots to prevent damage to the optics. Therefore, the microscope objective does not alter the laser intensity on target. We have clarified this point in the revised manuscript.

6. Figure 2 in the Methods section is not referenced anywhere in the main text. In addition, the spatial scale should be indicated on the image.

Our response 6:

Thank you for pointing this out. Figure 2 in the original manuscript was intended to be referenced in the subsection “*Justification criteria of high-quality shots*” within the METHODS section. Due to a typo, it was mistakenly referred to as Figure 6. This has been corrected in the revised manuscript, and the spatial scale has now been added to the image and clarified in the figure caption.